# Current pattern of antibiotic resistance and molecular characterization of virulence genes in *Klebsiella pneumoniae* obtained from urinary tract infection (UTIs) patients, Peshawar

Zeeshan Khan[1], Qaisar Ali[1], Sadiq Azam[1]*, Ibrar Khan[1], Jamila Javed[2], Noor Rehman[3], Mesaik M. Ahmed[4,5], Jalal Uddin[6], Ajmal Khan[7,8], Ahmed Al-Harrasi[7]*

1 Center of Biotechnology and Microbiology, University of Peshawar, Peshawar, Khyber Pakhtunkhwa, Pakistan, 2 Institute of Biotechnology Genetic Engineering, The University of Agriculture, Peshawar, Khyber Pakhtunkhwa, Pakistan, 3 Department of Pathology, Khyber Teaching Hospital Peshawar, Peshawar, Khyber Pakhtunkhwa, Pakistan, 4 Department of Medical Microbiology, Faculty of Medicine, University of Tabuk, Tabuk, Saudi Arabia, 5 Department of Medical Microbiology, Molecular Microbiology and Infectious Diseases Unit, Faculty of Medicine, University of Tabuk, Tabuk, Saudi Arabia, 6 Department of Pharmaceutical Chemistry, College of Pharmacy, King Khalid University, Abha, Kingdom of Saudi Arabia, 7 Natural and Medical Sciences Research Center, University of Nizwa, Birkat Al Mauz, Nizwa, Sultanate of Oman, 8 Department of Chemical and Biological Engineering, College of Engineering, Korea University, Seoul, Republic of Korea

* drazam@uop.edu.pk (SA); aharrasi@unizwa.edu.om (AA-H)

## Abstract

The current study investigates the prevalence of virulence genes obtained from clinical isolates of multidrug-resistant (MDR) *Klebsiella pneumoniae* at Khyber Teaching Hospital Peshawar, from October 2021 to January 2023. Upon proper consent, clinical samples of suspected UTIs patients were collected and inoculated on the nutrients agar media, McConkey agar media, and Cysteine Lysine Electrolyte Deficient (CLED) agar media followed by incubation at 37°C for 24 hrs. The phenotypic and genotypic identification were employed for the bacterial isolates. The phenotypic identification includes gram staining followed by the Analytical Profile Index (API 20E). A total of 215 (3.85%) positive isolates were found with the highest prevalence observed among the female patients (4.35%) followed by male (3.26%). The highest prevalence, constituting 52.55% (n=113), was detected in the age group of 21-40 years, followed by 31.62% (n=68) in the 41-60 age group. Additionally, 10.23% (n=22), 3.25% (n=7), and 2.32% (n=5) of cases were identified in the age groups of 01-10 years, 11-20 years, and above 60 years, respectively. Among the total positive samples, 44.65% (n=96) were collected from the Outpatient department (OPD), while inpatient department (IPD) cases contributed 55.35% (n=119). The antibiotic susceptibility profile of *K. pneumoniae* showed significant resistance to trimethoprim/Sulfamethoxazole (93%) and Colistin (79.07%). Tigecycline emerged as the most effective antibiotic with a sensitivity rate of 90%, along with Cefepime at the same level. Minimum Inhibitory Concentration (MIC) values indicated higher resistance for CTX, MEM, CN, AK, DO, CIP, and SXT in *K. pneumoniae-causing* UTIs from KTH, Peshawar.

**Data availability statement:** All relevant data are within the paper and its Supporting Information files.

**Funding:** The authors extend their appreciation to the Deanship of Scientific Research at King Khalid University for funding this work through the Large Groups Project under grant number (RGP2/98/45). The project was supported by grant from The Oman Research Council (TRC) through the funded project (BFP/RGP/HSS/23/037).

**Competing interests:** The authors have declared that no competing interests exist.

Molecular characterization of virulence genes reveals the highest prevalence of *fimH* (80%) followed by *SAT* (65%), *papEF* (49%), *afa* (29%), and *VAT* (16%). The sequencing data of the virulence genes reveals mutations in *fimH* and *papEF*, while *sat, afa* and *vat* virulence genes showed no mutations. The Chi-square test indicated a significant association between the types of bacteria, supporting our null hypothesis with a significance level of $p \leq 0.05$. The current study's finding is to evaluate the rise of antibiotic resistance in hospital settings, which highly demands the focus of health authorities and clinicians to manage the burden of the disease effectively.

## Introduction

*Klebsiella pneumoniae* is a Gram-negative, non-motile, facultative anaerobic bacterium in rod-shaped forms, varying in size from 1–2µm. It is commonly found in different strains as commensal, with some strains being pathogenic to humans [1]. Typically found in the human intestine, *K. pneumoniae* can cause various infections, including bacteremia, suppurative infections, soft tissue infections, osteomyelitis, or meningitis, and usually immunocompromised patients. Sometimes it colonizes human mucosal surfaces in the oropharynx and gastrointestinal (GI) tract [2]. *K. pneumoniae* is a member of the ESKAPE (*Enterococcus spp*, *staphylococcus aureus*, *Klebsiella pneumoniae, Acinetobacter baumannii*, *Pseudomonas aeruginosa*, and *Enterobacter spp*) group and is considered the second-most uropathogenic bacterium after *E. coli* [3]. This bacterial species is a significant challenge due to its MDR, as it is highly reported for producing β-lactamases. Such resistance leads to limited treatment options. However, the variation in the pathogenicity of the *K. pneumoniae* strains can be attributed to the presence or expression of the virulence factors [4]. The enhancement of *K. pneumoniaes's* pathogenicity is attributed to the presence of the MDR virulence Mega plasmid. Well-characterized virulence factors in *K. pneumoniae* include siderophores, lipopolysaccharides, and fimbriae [5]. These essential virulence factors are crucial in the initial stages of infection [6]. These structural components are critical in adherence, invasion, and development of infections [7]. In recent studies, several virulence factors, including porins, outer membrane proteins, iron transport systems, efflux pumps, and genes involved in allantoin metabolism, have been thoroughly characterized. Surface anchored proteins (SAT) are involved in the adhesion of bacterium to host cells [8]. Once attached to the host cell, Sat triggers the activation of signaling pathways that lead to the uptake of *K. pneumoniae* by the host cell. Type 1 fimbrial adhesin is found on the tip of fimbriae, which are hair-like structures that extend from the surface of *K. pneumoniae* [9]. The *fim*H is responsible for the binding of *K. pneumoniae* to specific carbohydrate molecules on the surface of host cells particularly in the urinary and respiratory tracts [9]. This interaction also ends in bacterial uptake. The absence of either Sat or *Fim*H significantly weakens *K. pneumoniae* virulence. Additionally, a multitude of factors, including the capsule, lipopolysaccharide, siderophores, and efflux pumps, contribute to the pathogen's virulence [10]. The presence of virulence factors and drug resistance mechanisms enhances the pathogen's capability to establish and persist in the colonized form. Therefore, considering the pathogenicity of *K. pneumoniae* strains, molecular characterization of virulence genes in both community and hospital settings is necessary for the development of potential antibiotic therapies. Different studies have been reported on the epidemiology of resistant genes, but limited research has addressed the genomics and pathogenicity of different strains [11]. Therefore, the current research work was conducted to assess antibiotic susceptibility testing and molecular characterization of virulence genes in clinical isolates of *K. pneumoniae* obtained from UTIs patients in KTH, Peshawar.

## Methodology

### Isolation and identification of clinical samples

The current research study was conducted in the pathology laboratory at KTH, Peshawar, and the Molecular and Genomics laboratory, at the Center of Biotechnology and Microbiology, University of Peshawar from October 2021 to January 2023. Upon proper consent, the patient's medical history, along with various parameters such as gender and age distribution. The study was approved by Khyber Medical College Peshawar Institutional Research and Ethical Board (IREB) Pakistan (No.124/DME/KMC).

Urine samples were collected from the UTIs patients and were inoculated on the nutrients agar media, McConkey agar media, and CLED agar media. The growth of bacterial samples was observed after incubation at 37°C for 24 hrs. Subsequently, both Phenotypic and genotypic identification were employed to identify the bacterial isolates. The phenotypic identification includes gram staining followed by the API 20E.

### Extraction of genomic DNA

Genomic DNA was extracted from the 24 h old culture of confirmed bacterial isolates using a Thermo Scientific DNA Purification Kit. A 1% agarose gel was prepared in 1X Tris Acetate EDTA (TAE) buffer, and gel electrophoresis was conducted to confirm the presence of DNA. Gel Doc™ Bio-Rad Molecular imager® was utilized to visualize the bands.

### Molecular identification of bacterial isolates

Genotypic identification via a specific primer of *WBBZ* (567bp) as shown in Table 2, under optimized conditions was performed for confirmation of the bacterial isolates. The specific gene was amplified, followed by electrophoresis on a 2% agarose Gel. The resulting bands were visualized by using the Gel Doc™ Bio-Rad Molecular imager®.

### Antibiotic susceptibility testing

Using the Kirby Bauer disc diffusion method, the antibiotic susceptibility pattern of bacterial isolates was determined against various groups of antibiotics following the guidelines of the Clinical and Laboratory Institutes (CLSI-2022). A standardized bacterial suspension equivalent to 0.5 McFarland turbidity was prepared from the isolate. The suspension was evenly spread across the surface of a sterile Muller Hinton agar (MHA) plate using a sterile cotton swab to create a uniform lawn of bacteria. The swab was passed in three directions to ensure consistent coverage, with a final sweep along the edge of the plate. The agar plate was allowed to dry for a few minutes at room temperature to ensure proper adherence to the bacterial inoculum. Next, selected antibiotic discs were carefully placed on the agar surface using sterile forceps, ensuring even spacing to avoid overlapping zones of inhibition. The plates were then incubated for 24 h at 37°C in an inverted position to prevent condensation from affecting the diffusion of antibiotics. After incubation, the zones of inhibition around each disc were measured in millimeters. The results were interpreted according to CLSI guidelines as intermediate (I), resistant (R), or sensitive (S) as shown in Table 1.

### Molecular characterization of the virulence genes

Specific PCR primers for the virulence genes as shown in Table 2, under optimized conditions, were employed in the investigation of all bacterial isolates. A total PCR reaction volume of 27 μl was prepared by combining 12.5 μl of 2x Thermo scientific master mix, 1 μl of forward and reverse primer, 2 μl of DNA template, and the remaining volume was filled with PCR grade

**Table 1. Selected specific antibiotics and their concentration used in the current research study.**

| S. No | Antibiotic (Symbol) | Concentration (µg) | Inhibition zone (mm) | | |
|---|---|---|---|---|---|
| | | | S | I | R |
| 1 | Amikacin (AK) | 30 | ≥30 | 15–16 | ≤14 |
| 2 | Gentamicin (CN) | 10 | ≥15 | 13–14 | ≤12 |
| 3 | Cefepime (FEP) | 30 | ≥25 | 19–24 | ≤18 |
| 4 | Amoxicillin-Clavulanate (AMC) | 10/20 | ≥18 | 14–17 | ≤13 |
| 5 | Levofloxacin (LEV) | 5 | ≥17 | 14–16 | ≤13 |
| 6 | Imipenem (IPM) | 10 | ≥23 | 20–22 | ≤19 |
| 7 | Colistin (CO) | 30 | ≥14 | 12–14 | ≤12 |
| 8 | Cefotaxime (CTX) | 30 | ≥26 | 23–25 | ≤22 |
| 9 | Piperacillin-tazobactam (TZP) | 10/100 | ≥21 | 18–20 | ≤17 |
| 10 | Ampicillin (AMP) | 10 | ≥17 | 14–16 | ≤13 |
| 11 | Sulfamethoxazole (SXT) | 12.5/23.75 | ≥16 | 11–15 | ≤10 |
| 12 | Ciprofloxacin (CIP) | 10 | ≥26 | 22–25 | ≤21 |
| 13 | Tobramycin (TOB) | 10 | ≥15 | 13–14 | ≤12 |
| 14 | Meropenem (MEM) | 10 | ≥23 | 20–22 | ≤19 |
| 15 | Cefoperazone-sulbactam (SCF) | 30/75 | ≥21 | 16–20 | ≤15 |
| 16 | Tigecycline (TGC) | 15 | ≥18 | 16–17 | ≤15 |

**Table 2. Specific primers sequence used for molecular identification and virulence genes.**

| Target Genes | Primer Sequence (5'–3') | Amplicon Size | PCR Conditions | References |
|---|---|---|---|---|
| *wbbz* | F: AGGATTGTATTCTGAAGGTC<br>R: TCAACTTGCCGTAATAAAGC | 576 | Annealing(°C/s): 53/30 | Present study |
| *Sat* | F: CTACAGCTTGATCACCTATGGC<br>R: CTCCCTGGTATTTCTTTGTGG | 410 | Annealing(°C/s): 59/60 | [12] |
| *Vat* | F: TTCACGGTACTGTTGTTCGC<br>R: CAGATAACTCCAGCGTCACG | 217 | Annealing(°C/s): 54/60 | [12] |
| *fimH* | F: CGGCGTGTTATCTAGTTTTTCC<br>R: TAGGTAATACCCCAGGTTTTGG | 397 | Annealing(°C/s): 57/60 | [13] |
| *pap*EF | F: GCAACAGCAACGCTGGTTGCATCAT<br>R: AGAGAGAGCCACTCTTATACGGACA | 336 | Annealing(°C/s): 57/30 | [14] |
| *Afa* | F: GCTGGGCAGCAAACTGATAACTCTC<br>R: CATCAAGCTGTTTGTTCGTCCGCCG | 750 | Annealing(°C/s): 57/30 | [14] |

water. The amplified products were then run on 2% agarose gel and visualized using the Del Doc® system.

## Minimum inhibitory concentration (MIC)

The effectiveness of antibiotics is determined by their MIC values by using E-strips as shown in Table 3. The clinical isolates were inoculated on MHA media and strips were employed followed by incubation at 37°C for 24 h.

## Mutational analysis of PCR products

The amplified DNA was sent to the Genomic and Sequencing Laboratory of Khyber Medical University, Peshawar for sequencing by using the Sanger Sequencing method. The sequencing data were analyzed by different bioinformatics tools such as Bio-edit sequence alignment

**Table 3. E-Strip used for the MIC determination.**

| S. NO | Antibiotic | MIC E-Strips | Symbol | Breakpoint | | |
|---|---|---|---|---|---|---|
| | | | | S | I | R |
| 1 | Cefotaxime | E-CT | CTX | ≤1 | 2 | ≥4 |
| 2 | Meropenem | E-MP | MEM | ≤1 | 2 | ≥4 |
| 3 | Gentamycin | E-GM | CN | ≤4 | 8 | ≥16 |
| 4 | Doxycycline | E-DC | DO | ≤4 | 8 | ≥16 |
| 5 | Co-Trimoxazole | E-TS | SXT | ≤2/38 | – | ≥4/76 |
| 6 | Amikacin | E-AK | AK | ≤16 | – | 64 |
| 7 | Ciprofloxacin | E-CL | CIP | ≤0.25 | 0.5 | ≥1 |

editor, and CLUSTLW. The consensus sequences were generated for each gene and checked through the NCBI Basic Local Alignment Search Tool (BLAST) to determine the local similarity between them.

## Statistical analysis

An analysis using SPSS version 20 utilized chi-square to determine the association between the expected value and the observed value, revealing a significance level of $p \leq 0.05$. This analysis was conducted with a sample size (n) of 150, employing degrees of freedom calculated as n-1.

## Results

### Isolation and identification of *Klebsiella pneumoniae*

A total of 5,580 clinical isolates were analyzed, of which 215 were detected positive for *K. pneumoniae,* isolated in KTH, Peshawar from Urine samples. The highest prevalence, constituting 52.55% (n = 113), was detected in the age group of 21–40 years, followed by 31.62% (n = 68) in the 41-60 years age group. Additionally, 10.23% (n = 22), 3.25% (n = 7), and 2.32% (n = 5) of cases were identified in the age groups of 01–10 years, 11-20 years, and above 60 years, respectively. Among the total positive samples, 44.65% (n = 96) were collected from the Outpatient department (OPD), while inpatient department (IPD) cases contributed 55.35% (n = 119) as shown in Table 4. For the precise and accurate identification of *K. pneumoniaee* clinical isolates, the API kit was employed as shown in Fig 1 and S3 Fig. The API system includes a series of biochemical tests designed to exploit the metabolic activities of bacteria. Positive results were observed for catalase and urease activity, demonstrating the organism's ability to hydrolyze urea and produce ammonia. Additionally, *K. pneumoniaee* exhibited positive fermentation of glucose and lactose, with acid detected as the product. Other biochemical tests, including indole production, phenylalanine deaminase activity, sucrose fermentation, and citrate utilization, also yielded positive results, further confirming the identity of the isolate.

### Antibiotic susceptibility testing

The antibiotic-sensitivity pattern of *K. pneumoniae* revealed the resistance-sensitive percentage values in the Table 2. Especially, SXT stands out with the highest resistance rate, reaching 93%. Following closely is CO, exhibiting a resistance rate of 79.07%. Especially, SXT stands out with the highest resistance rate, reaching 93%. Following closely is CO, exhibiting a resistance rate of 79.07%. The TGC emerges as the most effective antibiotic, demonstrating a

**Table 4. Different parameters of the clinical isolates obtained from UTIs patients.**

| Parameters | | *Klebsiella pneumoniae* (n = 215) |
|---|---|---|
| | | Number (%) |
| Gender | Female | 131 (4.35%) |
| | Male | 84 (3.26%) |
| Different age groups | 01–10 yrs | 22 (10.2%) |
| | 11–20 yrs | 7 (3.2%) |
| | 21–40 yrs | 113 (52.5%) |
| | Above 60 yrs | 68 (31.6%) |
| Patients' status | OPD | 96 (44.6%) |
| | IPD | 119 (55.3%) |

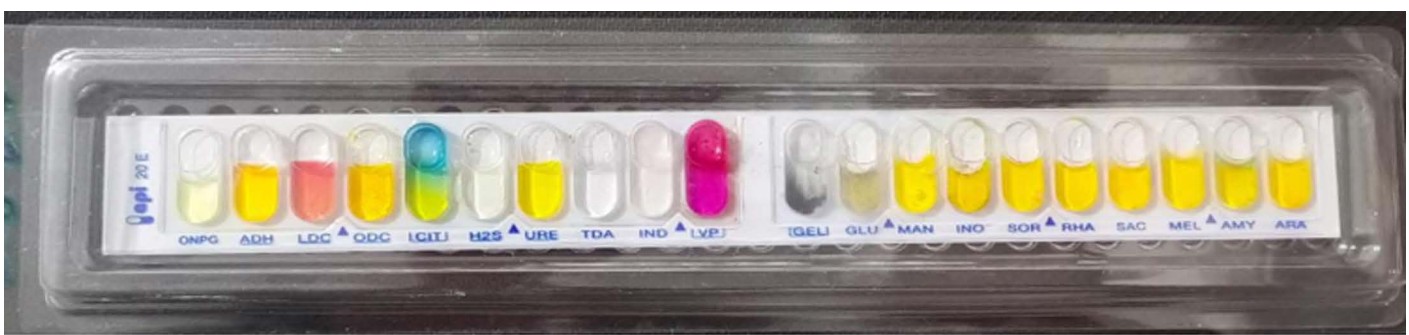

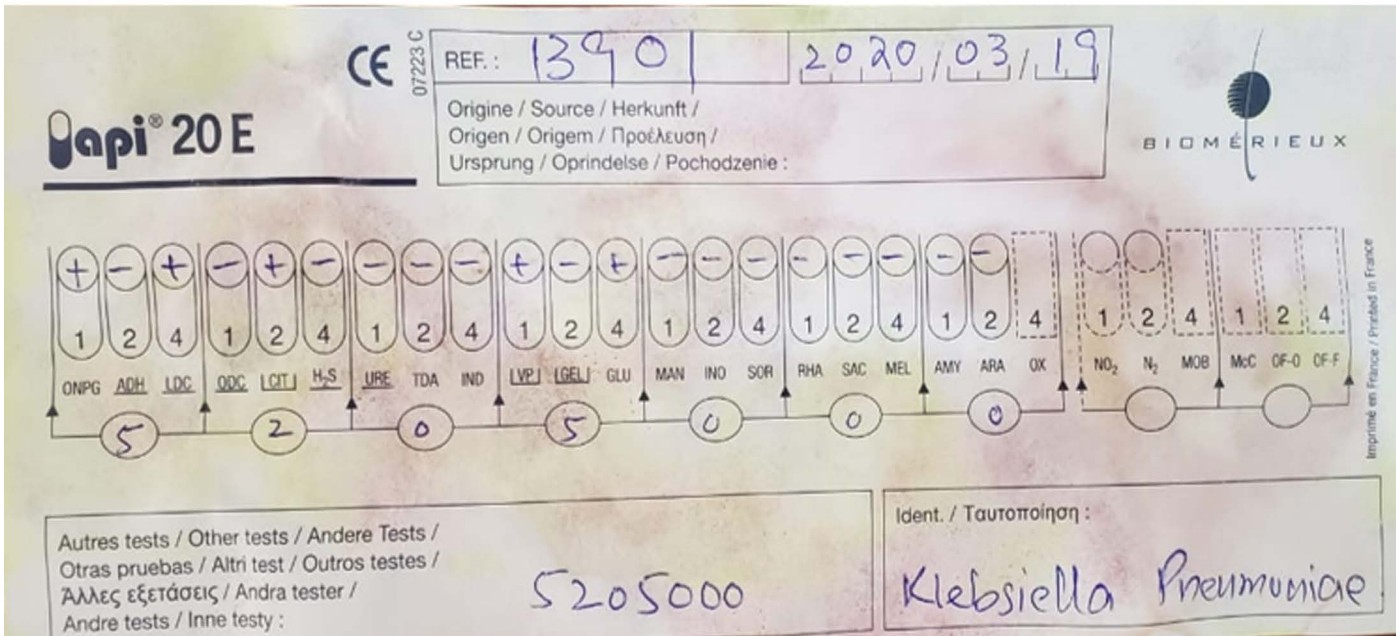

**Fig 1. Analytical profile index result for the identification of *Klebsiella pneumoniae*.**

sensitivity rate of 90%. Similarly, FEP shows a high sensitivity level, also standing at 90%. The overall values are presented in Table 5.

## Minimum inhibitory concentration (MIC)

The potency of antibiotics can be determined by their MIC valves, with MIC valves signifying greater effectiveness and higher MIC valves indicating diminished potency (S1, S2, S5 and S6 Figs). The *K. pneumoniae* causing UTIs obtained from the KTH, Peshawar revealed more resistance in the case of CTX, MEM, CN, AK, DO, CIP, and SXT as shown in Table 6.

## Comprehensive molecular characterization of virulence genes in *Klebsiella pneumoniae*

A detailed analysis of the virulence genes among the total isolates of *K. pneumoniae* reveals the highest prevalence of *fimH* (80%) followed by *SAT* (65%), *papEF* (49%), and *afa* (29%). The lowest prevalence of *VAT* (16%) was recorded as shown in Table 7 and Fig 2 and S4 and S7 Figs.

## Combination of different virulence genes in *K. pneumoniae*

The distribution of virulence gene combinations is briefly presented, detailing the number of genes, occurrence frequency, and percentage for each combination. The most prevalent combination is *Sat/fimH/papEF* which is observed in 24 isolates accounting for 11.01% of the total occurrences. Following closely is Sat/fimH, with two genes occurring 21 times, constituting 9.5% of total isolates. Other distinguished combinations include *fimH/papEF* (2 genes, 15 occurrences, 6.9%), *Sat/fimH/afa* (3 genes, 11 occurrences, 5.19%), *Sat/fimH/afa/vat* (4 genes, 10 occurrences, 4.5%), *Sat/afa/vat* (3 genes, 9 occurrences, 4.34%), *Sat/fimH/papEF/vat* (5 genes, 6 occurrences, 2.9%), *Sat/fimH/papEF/afa* (4 genes, 7 occurrences, 3.32%), and *Sat/fimH/papEF/vat/afa* (5 genes, 6 occurrences, 2.7%) as shown in Table 8.

Table 5. Antibiotic susceptibility pattern of *K. pneumoniae* against selected antibiotics.

| S. No | Antibiotic (Symbol) | Resistance (%) | Sensitive (%) |
|---|---|---|---|
| 1 | Amikacin (AK) | 82.0 (38) | 133.0 (62) |
| 2 | Gentamicin (CN) | 76.0 (35) | 139.0 (65) |
| 3 | Cefepime (FEP) | 22.0 (10) | 193.0 (90) |
| 4 | Amoxicillin-Clavulanate (AMC) | 166.0 (87) | 49.0 (23) |
| 5 | Levofloxacin (LEV) | 153.0 (71) | 62.0 (29) |
| 6 | Imipenem (IPM) | 129.0 (60) | 86.0 (40) |
| 7 | Colistin (CO) | 170.0 (55) | 45.0 (20.9) |
| 8 | Cefotaxime (CTX) | 155.0 (72) | 60.0 (28) |
| 9 | Piperacillin-tazobactam (TZP) | 64.0 (30) | 151.0 (70) |
| 10 | Ampicillin (AMP) | 162.0 (75) | 53.0 (25) |
| 11 | Sulfamethoxazole (SXT) | 200.0 (93) | 15.0 (7) |
| 12 | Ciprofloxacin (CIP) | 162.0 (75) | 53.0 (25) |
| 13 | Tobramycin (TOB) | 65.0 (30) | 150.0 (70) |
| 14 | Meropenem (MEM) | 54.0 (25) | 161.0 (75) |
| 15 | Cefoperazone-sulbactam (SCF) | 78.0 (36) | 137.0 (64) |
| 16 | Tigecycline (TGC) | 22.0 (10) | 193.0 (90) |

**Table 6. MIC of the selected antibiotic used against *K. pneumoniae*.**

| Antibiotics | MIC$_{50}$ (µg/ml) | MIC$_{90}$ (µg/ml) | MIC range(µg/ml) |
|---|---|---|---|
| Cefotaxime (CTX) | 128 | 192 | 4–192 |
| Meropenem (MEM) | 4 | 24 | 3–24 |
| Gentamicin (CN) | 16 | 16 | 4–16 |
| Amikacin (AK) | 16 | 192 | 1–192 |
| Doxycycline (DO) | 16 | 192 | 1–192 |
| Ciprofloxacin (CIP) | 32 | 256 | 0.094–256 |
| Trimethoprim/Sulfamethoxazole (SXT) | 32 | 32 | 0.064–32 |

**Table 7. Distribution of virulence genes detected in *K. pneumoniae*.**

| Virulence genes | Positive isolates (%) | Negative isolates (%) |
|---|---|---|
| *fim*H | 172 (80) | 43 (20) |
| *SAT* | 140 (65) | 75 (35) |
| *pap*EF | 105 (49) | 110 (51) |
| *afa* | 63 (29) | 152 (71) |
| *VAT* | 35 (16) | 180 (84) |

## Distribution and prevalence of virulence gene combinations in clinical isolates

The analysis reveals a significant statistical correlation between antibiotic resistance phenotypes and the presence of virulence genes. The presence of different virotypes showed a statistically insignificant association (P value > 0.05) with antimicrobial resistance phenotypes. An inverse relationship was observed across all studied genes, virotypes, and phenotypic antibiotic resistance. The odds ratio indicated a negative correlation, with values less than 1. The various combinations of virulence genes and their corresponding association data are presented in Table 9.

## Statistical analysis

The Chi-square test indicated a significant association between the type of bacteria, supporting our null hypothesis with a significance level of p ≤ 0.05. The one-way ANOVA test demonstrated a noteworthy relationship between the dependent and independent variables.

## Discussion

Multidrug resistance *K. pneumoniae* is one of the leading causes of life-threatening among several infections globally. Ghafourian *et al.* reported that the extensive utilization of antimicrobial agents has contributed to the increase in the prevalence of MDR *K. pneumoniae* [15]. Ciccozzi *et al.* reported in the study that the rising prevalence of MDR in *K. pneumoniae* strains is a significant and determined public health risk, contributing to elevated levels of illness and death globally [16]. Manjula *et al.* reported that Gram-negative bacteria have developed mechanisms to resist existing antibiotics, highlighting the challenge in treating bacterial infections. Similarly, in the current research study, the highest prevalence rate of 89% MDR *K. pneumoniae* was identified, consistent with the 90.2% prevalence by Manjula *et al.* in the literature. In the reported study most of the strains showed resistance to various antibiotics including cephalosporins, penicillin, fluoroquinolones, sulfonamides, and

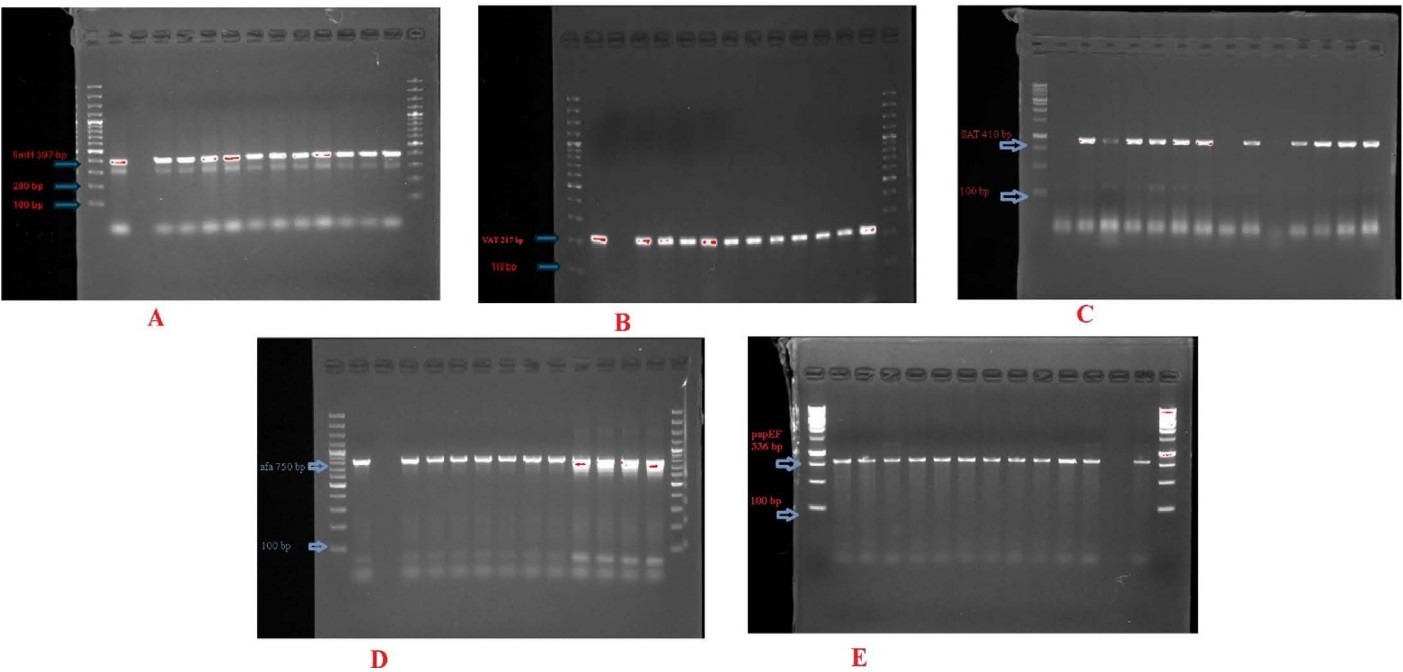

**Fig 2.** (A) L2-L7 showing PCR products of *FimH* (397 bp); (B) L2-L9 representing PCR products of *VAT* (217 bp); (C) DNA ladder of 100 bp in L1 and L8, with L2-L7 showing PCR products of *SAT* (410 bp); (D) L1 and L10 illustrating a 100 bp DNA ladder, with L2-L9 representing PCR products of the *afa* gene (750 bp), (E) DNA ladder of 100 bp in L1 and L9, with L2-L8 showing PCR products of *pap*EF (336 bp).

**Table 8.** Combination of the virulence genes and their frequency detected in *K. pneumoniae*.

| Gene Combination | Number of Genes | Occurrence Frequency | Percentage |
|---|---|---|---|
| Sat/fimH/papEF | 3 | 24 | 11.01 |
| Sat/fimH | 2 | 21 | 9.5 |
| fimH/papEF | 2 | 15 | 6.9 |
| Sat/fimH/afa | 3 | 11 | 5.19 |
| Sat/fimH/afa/vat | 4 | 10 | 4.5 |
| Sat/afa/vat | 3 | 9 | 4.34 |
| Sat/fimH/papEF/vat | 5 | 6 | 2.9 |
| Sat/fimH/papEF/afa | 4 | 7 | 3.32 |
| Sat/fimH/papEF/vat/afa | 5 | 6 | 2.7 |

aminoglycosides [17]. The antibiotic susceptibility profile of *K. pneumoniae* showed significant resistance to Amoxicillin/Clavulanate (87%), trimethoprim/Sulfamethoxazole (93%), and Colistin (79.07%). The reported study reveals that MDR *K. pneumoniae* is responsible for hospital-acquired infection and was found highly resistant toward tetracycline (95.2%), ciprofloxacin, and gentamycin (76.5% each), sulphathiazole (66.7%), nalidixic acid (61.9%) and norfloxacin (42.9%) [18]. Similarly, Indrajitha *et al.* (2021) reported a study regarding the drug resistance of *K. pneumoniae* which highlighted the 38% resistance to imipenem and 31% to meropenem respectively. The highest prevalence rate was detected in female patients (n = 131, 4.3%), followed by male patients (n = 84, 3.26%) in clinical isolates of *K. pneumoniae* in the current research study as supported by the reported study [19]. The prevalence

**Table 9. Combination of the virulence gene association with antibiotic association frequency.**

| Phenotypic Antibiotics Resistance Profile | | | | | | | | | | | | | | | |
|---|---|---|---|---|---|---|---|---|---|---|---|---|---|---|---|
| Virulence Genes | AMP | | | AMC | | SAM | | TZP | | FEP | | CN | | SXT | | C |
| | Total isolates | Number | Percentage | Number | Percentage | Number | Percentage | Number | Percentage | Number | Percentage | Number | Percentage | Number | Percentage | Number |
| Vat/sat | 23 | 23 | 100 | 23 | 100 | 6 | 26 | 13 | 56 | 16 | 69 | 13 | 56 | 17 | 73 | 6 |
| Sat/ fimH/ papEF | 4 | 3 | 75 | 3 | 75 | 1 | 25 | 4 | 100 | 3 | 75 | 2 | 50 | 3 | 75 | 1 |
| sat/ fimH | 26 | 26 | 100 | 15 | 57 | 3 | 11 | 15 | 57 | 21 | 80 | 0 | 00 | 24 | 92 | 4 |
| fimH/ papEF | 5 | 5 | 100 | 4 | 75 | 0 | 00 | 4 | 75 | 5 | 100 | 0 | 00 | 4 | 75 | 0 |
| Sat/ fimH/ afa | 3 | 3 | 100 | 3 | 100 | 0 | 00 | 1 | 25 | 3 | 75 | 1 | 25 | 2 | 66 | 1 |
| Sat/ fimH/ afa/vat | 1 | 1 | 100 | 0 | 0 | 0 | 00 | 0 | 00 | 1 | 100 | 0 | 00 | 1 | 100 | 0 |
| Sat/ afa/vat | 1 | 1 | 100 | 1 | 100 | 0 | 00 | 0 | 00 | 1 | 100 | 1 | 100 | 1 | 100 | 0 |
| sat/ fimH/ papEF/ vat | 3 | 3 | 100 | 1 | 33 | 0 | 00 | 2 | 66 | 2 | 67 | 0 | 0 | 3 | 100 | 0 |
| Sat/ fimH/ papEF/ afa | 19 | 19 | 100 | 15 | 78 | 5 | 26 | 10 | 52 | 15 | 78 | 0 | 00 | 16 | 84 | 6 |

of *K. pneumoniae* infection is increasing in Pakistan which is directly aligned with an increase in antibiotic resistance. This is consistent with the findings of Martin *et al.* (2016), who reported a 23% infection rate, and Zhang *et al.* (2018), who reported a 73.9% infection rate [20,21]. In the present study, the high prevalence rate of virulence genes such as *fim*H (80%), *SAT* (65%), *pap*EF (49%), *afa* (29%), and *VAT* (16%) in the MDR *K. pneumoniae* were observed. The strains obtained from patients with UTIs were investigated to determine their high potential rate of pathogenicity. A similar study of the virulence genes was reported in hospital-acquired infection caused by *K. pneumoniae* strains in China, the UK, France, and Brazil [22–25]. According to the reported study, Type-I fimbriae is the most common and frequent adhesive factor in the strains of *K. pneumoniae*. However, the presence of this gene has been associated with increased susceptibility to UTIs. This type 1 fimbrial adhesion has been identified as a mediating factor in the binding of *K. pneumoniae* strains to the mucous tissue layer of respiratory and Urinary tracts [26]. Wasfi *et al.* (2016) reported that most of the MDR strains of *K. pneumoniae* express type-I fimbrial adhesions. Similarly in the current research study, the prevalence of *FimH* (80%), *pap*EF (49%), and *afa* (29%), adhesive gene was identified [26]. Literature has reported that focused on the characterization of virulence factors in *K. pneumoniae* isolates performed statistical tests to evaluate the prevalence of *fimH*, which is crucial for adhesion and biofilm formation. The study utilized chi-square tests to analyze the correlation between the presence of *fimH* and the severity of UTIs, indicating a statistically significant association ($p \leq 0.05$) between *fimH* expression and increased virulence [27]. Another research paper assessed the prevalence of various virulence genes,

| Per-centage | FOS | | CTX | | CAZ | | ATM | | MEM | | IPM | | CN | | TOB | |
| --- | --- | --- | --- | --- | --- | --- | --- | --- | --- | --- | --- | --- | --- | --- | --- | --- |
| | Number | Per-centage | Number | Per-centage | Number | Per-centage | Number | Per-centage | Number | Per-centage | Number | Per-centage | Number | Per-centage | Number | Per-centage |
| 26 | 6 | 26 | 23 | 100 | 12 | 52 | 17 | 73 | 6 | 26 | 6 | 26 | 00 | 00 | 9 | 39 |
| 25 | 0 | 00 | 4 | 100 | 4 | 100 | 3 | 75 | 1 | 25 | 1 | 25 | 0 | 00 | 2 | 50 |
| 15 | 1 | 3.8 | 26 | 100 | 20 | 76 | 24 | 92 | 8 | 30 | 8 | 30 | 9 | 34 | 12 | 46 |
| 00 | 0 | 00 | 4 | 80 | 4 | 80 | 5 | 100 | 4 | 80 | 4 | 80 | 3 | 60 | 3 | 66 |
| 33 | 1 | 33 | 3 | 100 | 2 | 66 | 3 | 100 | 2 | 66 | 2 | 66 | 0 | 00 | 2 | 67 |
| 00 | 0 | 00 | 1 | 100 | 1 | 100 | 1 | 100 | 0 | 00 | 0 | 00 | 0 | 00 | 0 | 00 |
| 00 | 0 | 00 | 1 | 100 | 0 | 00 | 1 | 100 | 1 | 100 | 1 | 100 | 1 | 100 | 1 | 100 |
| 00 | 0 | 00 | 3 | 100 | 2 | 66 | 3 | 100 | 0 | 00 | 0 | 00 | 2 | 66 | 2 | 66 |
| 31 | 5 | 26 | 19 | 100 | 14 | 73 | 16 | 84 | 6 | 31 | 6 | 31 | 10 | 52 | 9 | 47 |

including *afa*, *vat*, *sat*, and *papEF*, among clinical isolates of *K. pneumoniae*. One-way ANOVA was employed to compare the mean expression levels of these genes in multidrug-resistant (MDR) and non-MDR strains. The results showed significant differences (p ≤ 0.05) in the expression of *vat* and *papEf* among different resistance profiles, revealing their potential role in the pathogenicity of MDR strain [28]. Similarly, in this current study, the virulence gene's association with antibiotic resistance was determined. The analyzed study statistically correlates antibiotic resistance phenotypes and virulence genes. The presence of various virotypes demonstrated a varied statistical connection (P value > 0.05) with antimicrobial resistance phenotypes. Overall, there was a noted inverse association observed among all studied genes, virotypes, and phenotypic antibiotic resistance. The odds ratio revealed a negative correlation, representing a value of less than 1. The finding and mutation in the virulence genes may offer a molecular explanation of antibiotic resistance observed in the isolates of the conducted study.

## Conclusion

The study highlights the critical global threat of antibiotic resistance, particularly in *K. pneumoniae*. The observed resistance in Gram-negative bacteria, remarkable to key antibiotics, signals a pressing need for alternative treatment strategies. The prevalence of MDR *K. pneumoniae* in hospital-acquired infections, especially among female patients, mirrors an alarming trend in Pakistan. The investigation into virulence gene expression provides valuable

awareness of the molecular basis of antibiotic resistance. Urgent and concerted efforts are needed to address this growing challenge and develop effective therapeutic approaches against MDR bacterial infections.

## Supporting information

**S1 Fig. Bacterial growth on MacConkey agar media**.
(JPG)

**S2 Fig. Gram staining of *K. Pneumoniae.***
(JPG)

**S3 Fig. Analytical Profile Index for the identification of *K. pneumoniae***.
(JPG)

**S4 Fig. Gel image of Wbbz (567 bp) for Molecular identification of *K. pneumoniae*: L1,L10: DNA ladder 100 bp, L2-L9: PCR product of Wbbz Gene.**
(JPG)

**S5 Fig. Representative image of Antibiogram of *K. pneumoniae.***
(JPG)

**S6 Fig. Representative image of determination of MIC of *K. pneumoniae* determined by E-strip.**
(JPG)

**S7 Fig. Original uncropped images of gel of Fig 2 of the main text.**
(DOCX)

## Acknowledgments

The authors would like to thank the Higher Education Commision (HEC), Pakistan and University of Nizwa (UoN) for the generous support of this project. Informed consent was obtained from all subjects involved in the study.

## Author contributions

**Conceptualization:** Qaisar Ali, Sadiq Azam, Ahmed Al-Harrasi.

**Data curation:** Zeeshan Khan, Qaisar Ali, Sadiq Azam, Ibrar Khan, Jamila Javed, Noor Rehman, Mesaik M. Ahmed, Ahmed Al-Harrasi.

**Formal analysis:** Zeeshan Khan, Qaisar Ali, Sadiq Azam, Noor Rehman, Mesaik M. Ahmed, Jalal Uddin, Ajmal Khan.

**Funding acquisition:** Jalal Uddin, Ajmal Khan, Ahmed Al-Harrasi.

**Investigation:** Sadiq Azam, Jamila Javed, Noor Rehman.

**Methodology:** Zeeshan Khan, Qaisar Ali.

**Project administration:** Ajmal Khan.

**Resources:** Jamila Javed, Ahmed Al-Harrasi.

**Software:** Jamila Javed.

**Supervision:** Sadiq Azam, Ahmed Al-Harrasi.

**Visualization:** Ibrar Khan, Noor Rehman, Mesaik M. Ahmed.

**Writing – original draft:** Zeeshan Khan, Qaisar Ali.

**Writing – review & editing:** Ibrar Khan, Mesaik M. Ahmed, Jalal Uddin, Ajmal Khan, Ahmed Al-Harrasi.

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
