## [Decision Letter · Decision Letter 0]

6 Dec 2024

PONE-D-24-44471Current Pattern of Antibiotic Resistance and Molecular Characterization of Virulence Gene in Klebsiella Pneumonia Obtained from Urinary Tract (UTIs) Infection Patient, PeshawarPLOS ONE

Dear Dr. Azam,

Thank you for submitting your manuscript to PLOS ONE. After careful consideration, we feel that it has merit but does not fully meet PLOS ONE’s publication criteria as it currently stands. Therefore, we invite you to submit a revised version of the manuscript that addresses the points raised during the review process.

Your manuscript has been reviewed and a minor revision is suggested. Please follow the comments and make all necessary revision. 

We look forward to receiving your revised manuscript.

Kind regards,

Yung-Fu Chang

Academic Editor

PLOS ONE

Journal requirements: When submitting your revision, we need you to address these additional requirements. 1. Please ensure that your manuscript meets PLOS ONE's style requirements, including those for file naming. The PLOS ONE style templates can be found at https://journals.plos.org/plosone/s/file?id=wjVg/PLOSOne_formatting_sample_main_body.pdf and https://journals.plos.org/plosone/s/file?id=ba62/PLOSOne_formatting_sample_title_authors_affiliations.pdf 2. Please amend either the title on the online submission form (via Edit Submission) or the title in the manuscript so that they are identical. 3. Please include a caption for table 2.  4. PLOS ONE now requires that authors provide the original uncropped and unadjusted images underlying all blot or gel results reported in a submission’s figures or Supporting Information files. This policy and the journal’s other requirements for blot/gel reporting and figure preparation are described in detail at https://journals.plos.org/plosone/s/figures#loc-blot-and-gel-reporting-requirements and https://journals.plos.org/plosone/s/figures#loc-preparing-figures-from-image-files. When you submit your revised manuscript, please ensure that your figures adhere fully to these guidelines and provide the original underlying images for all blot or gel data reported in your submission. See the following link for instructions on providing the original image data: https://journals.plos.org/plosone/s/figures#loc-original-images-for-blots-and-gels.   In your cover letter, please note whether your blot/gel image data are in Supporting Information or posted at a public data repository, provide the repository URL if relevant, and provide specific details as to which raw blot/gel images, if any, are not available. Email us at plosone@plos.org if you have any questions. 5. PLOS requires an ORCID iD for the corresponding author in Editorial Manager on papers submitted after December 6th, 2016. Please ensure that you have an ORCID iD and that it is validated in Editorial Manager. To do this, go to ‘Update my Information’ (in the upper left-hand corner of the main menu), and click on the Fetch/Validate link next to the ORCID field. This will take you to the ORCID site and allow you to create a new iD or authenticate a pre-existing iD in Editorial Manager. 6. Please ensure that you have specified a) Did participants provide their written or verbal informed consent to participate in this study?b) If consent was verbal, please explain i) why written consent was not obtained, ii) how you documented participant consent, and iii) whether the ethics committees/IRB approved this consent procedure." - In consent please state in Ethics Method section and manuscript if it is written or verbal. If consent was verbal, please explain a) why written consent was not obtained, b) how you documented participant consent, and c) whether the ethics committees/IRB approved this consent procedure. 7. Your ethics statement should only appear in the Methods section of your manuscript. If your ethics statement is written in any section besides the Methods, please move it to the Methods section and delete it from any other section. Please ensure that your ethics statement is included in your manuscript, as the ethics statement entered into the online submission form will not be published alongside your manuscript. 

Reviewers' comments:

Reviewer's Responses to Questions

**Comments to the Author**

1. Is the manuscript technically sound, and do the data support the conclusions?

Reviewer #1: Yes

2. Has the statistical analysis been performed appropriately and rigorously? 

Reviewer #1: No

3. Have the authors made all data underlying the findings in their manuscript fully available?

Reviewer #1: No

4. Is the manuscript presented in an intelligible fashion and written in standard English?

Reviewer #1: No

5. Review Comments to the Author

Reviewer #1: 1- All scientific name must be written in italic

2- Inoculation method in the antibiotics sensitivity test should be explained in more details.

3- Figures of gel electrophoresis should be with high resolution and the arrows pointed between two bands?!!

4- Statistical analysis not clearly showed in results section.

5- nutrient agar media not Nutrient agar Media.

6- The authors mentioned that bacterial isolates were identified by phenotype method using API test but the API test showed some biochemical not phenotype identification.

6. PLOS authors have the option to publish the peer review history of their article (what does this mean? ). If published, this will include your full peer review and any attached files.

**Do you want your identity to be public for this peer review?** For information about this choice, including consent withdrawal, please see our Privacy Policy .

Reviewer #1: **Yes: ** Hussein S. Salama

---

## [Author Response · Author response to Decision Letter 1]

8 Jan 2025

PLOS ONE

Response: We follow the guidelines and format the MS

2. Please amend either the title on the online submission form (via Edit Submission) or the title in the manuscript so that they are identical.

Response: Done

3. Please include a caption for Table 2.

Response: Done

Response: The original uncropped images of the gel are given in the supporting information.

Response: Done

6. Please ensure that you have specified a) Did participants provide their written or verbal informed consent to participate in this study?

b) If consent was verbal, please explain i) why written consent was not obtained, ii) how you documented participant consent, and iii) whether the ethics committees/IRB approved this consent procedure."

- In consent please state in the Ethics Method section and manuscript if it is written or verbal. If consent was verbal, please explain a) why written consent was not obtained, b) how you documented participant consent, and c) whether the ethics committees/IRB approved this consent procedure.

Response: Written consent has been taken.

7. Your ethics statement should only appear in the Methods section of your manuscript. If your ethics statement is written in any section besides the Methods, please move it to the Methods section and delete it from any other section. Please ensure that your ethics statement is included in your manuscript, as the ethics statement entered into the online submission form will not be published alongside your manuscript.

Response: done

8. Please review your reference list to ensure that it is complete and correct. If you have cited papers that have been retracted, please include the rationale for doing so in the manuscript text or remove these references and replace them with relevant current references. Any changes to the reference list should be mentioned in the rebuttal letter that accompanies your revised manuscript. If you need to cite a retracted article, indicate the article’s retracted status in the References list and also include a citation and full reference for the retraction notice.

Response: done

Reviewers' comments:

Reviewer's Responses to Questions

Comments to the Author

Reviewer #1: 1- All scientific names must be written in italic

Response: Corrected throughout in the MS

2- The inoculation method in the antibiotics sensitivity test should be explained in more detail.

Response: It is explained in detail and highlighted

3- Figures of gel electrophoresis should be with high resolution and the arrows pointed between two bands?!!

Response: A clear and high-resolution figure has been added.

4- Statistical analysis is not clearly shown in the results section.

Response: the correction has addressed

5- nutrient agar media, not Nutrient agar Media.

Response: Corrected

6- The authors mentioned that bacterial isolates were identified by phenotype method using the API test, but the API test showed some biochemical not phenotype identification.

Response: The correction has been addressed in accordance with the reviewer’s query.

---

## [Editor Report · Decision Letter 1]

30 Jan 2025

Current Pattern of Antibiotic Resistance and Molecular Characterization of Virulence Gene in Klebsiella pneumoniae Obtained from Urinary Tract (UTIs) Infection Patient, Peshawar

PONE-D-24-44471R1

Dear Dr. Azam,

We’re pleased to inform you that your manuscript has been judged scientifically suitable for publication and will be formally accepted for publication once it meets all outstanding technical requirements.

Kind regards,

Yung-Fu Chang

Academic Editor

PLOS ONE
---

## [Editor Report · Acceptance letter]

PONE-D-24-44471R1

PLOS ONE

Dear Dr. Azam,

I'm pleased to inform you that your manuscript has been deemed suitable for publication in PLOS ONE. Congratulations! Your manuscript is now being handed over to our production team.

Kind regards,

on behalf of

Dr. Yung-Fu Chang

Academic Editor

PLOS ONE